# Carboxylated Carbon Nanotube/Polyimide Films with Low Thermal Expansion Coefficient and Excellent Mechanical Properties

**DOI:** 10.3390/polym14214565

**Published:** 2022-10-27

**Authors:** Cheng Lu, Fangbing Lin, Huiqi Shao, Siyi Bi, Nanliang Chen, Guangwei Shao, Jinhua Jiang

**Affiliations:** 1Shanghai Frontier Science Research Center for Modern Textiles, College of Textiles, Donghua University, Shanghai 201620, China; 2Engineering Research Center of Technical Textiles, Ministry of Education, Donghua University, Shanghai 201620, China; 3Innovation Center for Textile Science and Technology, Donghua University, Shanghai 200051, China

**Keywords:** multiwalled carbon nanotubes, acid treatment, polyimide film, physical-chemical properties

## Abstract

Polyimide (PI) films with excellent heat resistance and outstanding mechanical properties have been widely researched in microelectronics and aerospace fields. However, most PI films can only be used under ordinary conditions due to their instability of dimension. The fabrication of multifunctional PI films for harsh conditions is still a challenge. Herein, flexible, low coefficient of thermal expansion (CTE) and improved mechanical properties films modified by carboxylated carbon nanotube (C-CNT) were fabricated. Acid treatment was adapted to adjust the surface characteristics by using a mixture of concentrated H_2_SO_4_/HNO_3_ solution to introduce carboxyl groups on the surface and improve the interfacial performance between the CNT and matrix. Moreover, different C-CNT concentrations of 0, 1, 3, 5, 7, and 9 wt.% were synthesized to use for the PI film fabrication. The results demonstrated that the 9 wt.% and 5 wt.% C-CNT/PI films possessed the lowest CTE value and the highest mechanical properties. In addition, the thermal stability of the C-CNT/PI films was improved, making them promising applications in precise and harsh environments.

## 1. Introduction

With the fast development of material science, the fields of aerospace, electronic engineering, communication, and architecture have undergone rapid changes, which put forward higher requirements for applied materials to have a variety of superior properties, such as improved mechanical properties and thermal stability. Therefore, functional polymers with superlative comprehensive properties have attracted more and more attention [1]. The density of polymer materials is far lower than that of metals and ceramics, and with the development of material technology, their other properties have also been greatly improved. Among these polymer materials, polyimide (PI) film materials have outstanding properties [2,3,4,5,6,7,8]: low density, high mechanical properties, great thermal stability, radiation resistance, corrosion resistance, excellent dielectric properties, and so on. Therefore, PI films can be used in microelectronics, aerospace, and other fields [9,10,11,12]. However, the coefficient of thermal expansion (CTE) of polyimide is much higher than that of inorganic materials such as metals and ceramics. Composites made of polyimide and inorganic materials will warp, crack or delaminate at high temperatures due to their mismatched CTE [13,14,15,16]. In addition, the mechanical properties of polyimide films are also insufficient in some applications requiring high strength [17]. Therefore, high-strength and low-CTE PI films are required to achieve applications in areas with higher performance requirements.

In recent years, carbon nanotubes (CNT) have begun to be applied to polymer composites. Since the advent of CNT [18], more and more scholars have studied them. CNT has a high aspect ratio, thermal conductivity, thermal stability, conductivity, and unique mechanical properties [19,20]. CNT can improve the thermal properties, mechanical properties, and other properties of polymer materials, with a significant enhancement even at very low concentrations (0.5 vol% [21], 0.79–5 wt.% [22,23,24,25,26,27,28,29,30]). Therefore, CNT has been well applied in aerospace and other fields. For example, aluminum was replaced by carbon nanotube fiber reinforced polymer (CNRP) on the Reusable Launch Vehicle to reduce the payload, leading to a weight reduction of 82% [31]. Adding CNT to the pure resin used by satellites can obtain better mechanical properties and thermal stability [32]. Poly (ionic liquid) s (PILs), as a critical material in the new generation of intelligent electromechanical equipment, has improved its electrical and mechanical properties after adding CNT [33]. Ju et al. [34] reported the fabrication of multiwalled CNTs/polyacrylonitrile fibers by electrospinning, which exhibited an electrical conductivity of 96 S/m at a nanotube loading of 3 wt.%. Jiang et al. [35] prepared multiwalled CNTs/PI composites, including 3,3′,4,4′-biphenyltetracarboxylic dianhydride (BPDA), p-phenylenediamine (p-PDA), which showed the effective absorption bandwidth of 2.72 GHz with the matching thickness of only 2.0 mm when the content of CNTs was 6 wt.%, and the tensile modulus and tensile strength of the composites all reached their maximum values of 4.9 GPa and 281.3 MPa when the content of CNTs was 0.5 wt.%, respectively.

In general, CNT can improve the thermal and mechanical properties of polymer materials. However, the main problems encountered in the preparation of CNT-reinforced composites are the aggregation and the weak interface interaction between CNT and the matrix with a high CNT content [36,37,38]. Self-organization of carbon nanotubes in the matrix can be realized through proper surface engineering, thus utilizing the aggregation of carbon nanotubes [39,40]. This controlled aggregation in a series of CNT systems can achieve splendid optoelectronic properties and promote the development of new nanotube sensors [41,42]. However, in the physical or chemical method commonly used to mix CNT and polymer, the phenomenon of agglomeration will lead to the deterioration of the dispersion of CNT, thus leading to the degradation of the properties of composites with high content of CNT. Therefore, the functionalization of CNT has been studied [43,44,45,46], and the surface chemical modification of CNT is used to reduce agglomeration. Zhang et al. [47] prepared acid-treated single-walled carbon nanotubes/PI films, which exhibited the best gas separation performance than the single-walled carbon nanotubes/PI composites without acid treatment at a filler content of 2 wt.%. Lee et al. [48] reported the unmodified, oxidized, and silanized CNTs/epoxy composites, the elastic modulus of the silanized CNTs/epoxy composite was 34% higher than that of the unmodified CNTs/epoxy composites, and 18% higher than that of the oxidized CNTs/epoxy composites. The surface functionalized CNT has improved dispersion in organic solvents and resin matrix, and the high dispersion of CNT can enhance the performance of CNT/PI films, thus giving CNT/PI films a broader application prospect.

Here, CNT was functionalized by acid treatment to improve its dispersion in polyamic acid (PAA) solution and PI film; thus, carboxylated CNT (C-CNT) was achieved. Next, CNT/PI and C-CNT/PI films were prepared with different contents of CNT and C-CNT, respectively. The dispersion of CNT and C-CNT, as well as the effects of filler content on the thermal and mechanical properties of the films, were systematically studied. Moreover, the thermal and mechanical performance differences between CNT/PI and C-CNT/PI films were compared under the same filler content.

## 2. Materials and Method

### 2.1. Materials

Polyamic acid (PAA, cas No. 25036-53-7, >99.6 wt.%, synthesized by 1,2,4,5-benzenetetracarboxylic anhydride (PMDA) and 4,4′-Oxydianiline (ODA)) was purchased from Xinsheng Plastic Company (Dongguan, China), which was dissolved in NMP at a ratio of 18 wt.%. N-methylpyrrolidone (NMP, cas No. 872-50-4, 99.5 wt.%) was obtained from Shanghai Adamas Reagents Company (Shanghai, China). Sulfuric acid (H_2_SO_4_, cas No. 7664-93-9, 95–98 wt.%) and nitric acid (HNO_3_, cas No. 7697-37-2, 65–68 wt.%) were purchased from Sinopharm Chemical Reagent Company (Shanghai, China). Multiwalled carbon nanotube (cas No. 308068-56-6, 95 wt.%, length 10–30 μm, diameter 10–20 nm) was purchased from Jiangsu XFNANO Materials Technique Company (Nanjing, China).

### 2.2. Preparation of C-CNT

C-CNT was prepared by acid treatment; 5 g CNT was dispersed in 400 mL of a mixture of concentrated H_2_SO_4_/HNO_3_ solution with a volume ratio of 3:1 and stirred at 50 °C for 20 h [45]. The product was filtered and collected through 0.45 μm pore-sized PTFE film. Afterward, the product was washed with deionized water and dried at room temperature for 12 h, repeated three times. Finally, ground the obtained C-CNT into powder and screened with a 280-mesh sieve (0.055 mm).

### 2.3. Preparation of CNT/PI and C-CNT/PI Films

The preparation of C-CNT/PI films consisted of two steps, as shown in Figure 1. In the first step, the predetermined amount of C-CNT was added to the NMP solution and dispersed in the following steps: stirred for 30 min, sonicated for 30 min, stirred for 6 h, and sonicated for 30 min. Then the C-CNT suspension was added to the PAA solution and dispersed by the same stirring and ultrasonic steps as above. At this time, all PAA solutions were diluted to 10 wt.% by NMP. In the second step, the C-CNT/PAA suspensions were cast onto the glass plate and evacuated for 30 min for degassing (≤−0.1 MPa). Then the CNT/PI films were prepared by thermal imidization at the following temperatures: 80 °C, 120 °C, 160 °C, 200 °C, 250 °C, 300 °C, all times were 1 h for each temperature, and finally cooled to room temperature (25–30 °C). Through this process, a series of C-CNT/PI films with C-CNT concentrations of 0, 1, 3, 5, 7, and 9 wt.% were synthesized. The average thickness of the films was about 80 μm. The preparation process for CNT/PI films was the same as the above steps.

### 2.4. Characterizations

Fourier transform infrared (FTIR) spectra of CNT and C-CNT were studied with a Fourier infrared spectrometer (FTIR, Nicolet6700, Benton, AR, USA) to detect any new absorption bands during the acid treatment. For sample preparation, the infrared KBr tableting method was applied. The spectrums were recorded with a resolution of 2 cm^−1^ in the range of 400–4000 cm^−1^. The surface chemistry of CNT and C-CNT was monitored by X-ray photoelectron spectroscopy (XPS, ESCALAB 250, Richardson, TX, USA). The pass energy was set at resolution 20 eV, and the X-ray source was Al-Ka (1486.6 eV). The water contact Angle of the film surface was measured by an optical surface analyzer (OSA200, Ningbo Scientific Instruments Company, Ningbo, China). The morphology of CNT and C-CNT in NMP solution was analyzed by a transmission electron microscope (TEM, JEM 2100F, Tokyo, Japan). The transmittance spectra of CNT/PI and C-CNT/PI films were measured by a UV-visible spectrophotometer (UV950, JASCO INTERNATIONAL Co., Ltd., Tokyo, Japan). The surface morphologies of films were observed by a scanning electron microscope (SEM, TM3000, Tokyo, Japan). Thermal gravimetric analysis (TGA, TGA 8000, New York, NY, USA) was used for thermal analysis of the CNT/PI films and C-CNT/PI films under a nitrogen atmosphere. The test samples were heated from 50 °C to 800 °C at the rate of 10 °C/min. CTEs of the films were performed on thermo-mechanical analysis (TMA, TMAQ400, Chicago, IL, USA) with a tension force of 0.05 N. The sample spacing was 16 mm, the temperature range was −10–150 °C, and the heating rate was 10 °C/min. Mechanical properties were tested by a microcomputer control electron universal testing machine (MTS, MTS-E42.503, Xiamen, China). The sample size was 100 mm × 10 mm, the gauge length was 50 mm, and the test was conducted at the crosshead speed of 5 mm/min. At least five samples were tested to analyze the mechanical properties of the films.

## 3. Results and Discussion

### 3.1. Chemical Composition of CNT and C-CNT

Initially, the performance of CNT and C-CNT was compared and analyzed. The functional group difference between CNT and C-CNT was investigated by FTIR spectroscopy, as shown in Figure 2. There were features at wavenumbers 1000–1200 cm^−1^, 1600–1680 cm^−1^, and 2840–3000 cm^−1^ for CNT and C-CNT, corresponding to C–O, C=C, –CH stretching, respectively. The features at wavenumbers 1406 cm^−1^ for CNT and 1455 cm^−1^ for C-CNT were attributed to –CH bend vibration. Compared with CNT, C-CNT showed additional characteristic peaks at 1164 cm^−1^ and 1735 cm^−1^, which can be attributed to the C–O (1000–1200 cm^−1^) and C=O (1650–1800 cm^−1^) stretching vibrations of the carboxylic group. Moreover, the peak observed for C-CNT at 3434 cm^−1^ originated from the stretching frequency of –OH (3200–3500 cm^−1^). The results show that –COOH and more –OH functional groups appeared on the surface of C-CNT compared with CNT.

XPS analysis was used to study the differences between elements and functional groups on the surface of CNT and C-CNT. Figure 3 presents C1s XPS spectra and XPS survey scans of CNT and C-CNT. The C1s peak was deconvoluted by XPS-PEAK software and divided according to four binding energy positions in order to analyze the content of different functional groups, as shown below: Sp^2^ C (284.8 eV), Sp^3^ C (285.1 eV), –C–O– (286.2 eV) and –CO–O– (290.1 eV). The content of the functional groups and element content of CNT and C-CNT were summarized in Table 1 and Table 2, respectively. More O atom content was observed in C-CNT, with a twofold increase in the O/C ratio compared to CNT, implying that C-CNT had more oxygen-containing functional groups on the surface. Compared with CNT, the content of –C–O– of C-CNT increased from 7.04% to 11.74%, and the content of –CO–O– also increased, which indicated that C-CNT had more –COOH and –OH functional groups, which was also in agreement with the results of FTIR analysis.

The contact angle can provide information about surface chemistry because it is sensitive to surface chemistry. Therefore, the water contact angle of the films was tested and summarized in Figure 4 and Table 3. The contact angle of CNT/PI and C-CNT/PI films increased with the increase in CNT and C-CNT content, and CNT/PI films exhibited larger contact angles than CNT/PI films. The main chain of polyimide contains carbonyl groups, which can form hydrogen bonds with hydroxyl groups, so the hydrophilicity was good (55.74°). As a super hydrophobic material with a water contact angle of up to 158° [49], CNT can improve the hydrophobicity of PI films. When the CNT content was 9 wt.%, the contact angle of CNT-PI film was increased to 81.17°, which is 45.62% higher than that of pure PI film. However, C-CNT was not so effective in improving the contact angle of PI films because the carboxyl group in C-CNT was hydrophilic, so the hydrophobicity of C-CNT was not as good as that of CNT [50]. As a result, the contact angle of C-CNT/PI films only increased to 76.23 when the content of C-CNT was 9%, which was 36.76% higher than that of pure PI films. In a word, CNT and C-CNT can improve the contact angle and hydrophobicity of PI films, and the effect is best when 9 wt.% CNT is added. As a result, the self-cleaning, antifouling, and anti-corrosion properties of the films have been improved, which has a broader prospect in fields such as aerospace and medical care.

### 3.2. Morphology and Dispersion of CNT and C-CNT in Solution and Film

TEM examination of CNT and C-CNT was performed to investigate the morphology and dispersion. For the CNT and C-CNT showed in Figure 5, the average diameter was in the range of 10–20 nm and had similar profiles. However, C-CNT with a shorter length appeared (at the white arrow in Figure 5d). Notably, CNT was usually closed-ended (indicated by the circles in Figure 5c), whereas C-CNT was typically open-ended (indicated by the circles in Figure 5d) [51]. The reason is that the reaction usually starts at the defect sites such as the heptatomic rings, the –CH_2_ and –CH groups [46], which often exist at both ends and surface of CNT, leading to the cutting and opening of carbon nanotubes after strong acid treatment. The dispersion of CNT and C-CNT in the NMP solution, as seen by TEM, was very different; CNT was clustered into bundles (Figure 5c), while C-CNT was randomly and loosely dispersed in the NMP solution (Figure 5d), neither of them had any particulate impurities. Due to the tiny size of CNT, there is a relatively strong van der Waals force, which makes it easy to become entangled or reunited into bundles. After acid treatment, C-CNT was cut short, reducing the agglomeration and bonding state of long fibers, and the carboxyl group on the surface of C-CNT increases the surface polarity of C-CNT, which makes C-CNT well dispersed in a polar solvent (NMP solution). In addition, C-CNT was observed to be locally agglomerated and present a buckling shape. This phenomenon can be related to them being mostly open-ended anchoring near CNTs because, after acid treatment, C-CNT was grafted with carboxyl groups (both ends and weak points on the surface), and hydrogen bonds were easily formed between carboxyl groups. Moreover, the structure of C-CNT was not damaged because both CNT and C-CNT showed a multiwall structure. In conclusion, C-CNT was grafted with new functional groups to open the fracture, and the tube wall structure was not damaged.

The dispersion of CNT and C-CNT in 10 wt.% PAA/NMP solutions is shown in Figure 6. It can be seen that the dispersion of C-CNT was much better than that of CNT. After standing for 36 h, the CNT suspensions containing 3 wt.% and 9 wt.% CNT began to settle, while C-CNT suspensions were still well dispersed. Even after standing for 72 h, 3 wt.% C-CNT suspensions had not entirely settled. This is due to the fact that –COOH groups on the surface of C-CNT belong to polar groups and are readily soluble in polar NMP solvents. In the process of curing the solution into a film, the great dispersion and stability of C-CNT in the solution make it maintain great dispersion in the film.

In order to analyze the influence of filler content on the apparent morphology of films, photographs of PI films filled with CNT or C-CNT with different contents were taken. As shown in Figure 7a,b, with the increase in CNT or C-CNT content, the color of the films gradually turned black. The surfaces of CNT/PI films and C-CNT/PI films were extremely smooth, within the range of 1–5 wt.% filler content. However, with the increase in filler content, at 9 wt.%, the surface of C-CNT/PI film was still smooth (Figure 7e), while the surface of CNT/PI film had visible particles (Figure 7d), which was due to the agglomeration of CNT. The behavior of CNT agglomeration can also be observed inside the film with high filler content [39,48]. Additionally, the film still exhibited good flexibility after adding filler, as shown in 9 wt.% C-CNT/PI film (Figure 7c).

With UV-visible spectroscopy, it is possible to detect an absorption band caused by the electron transfer (C_m_^1^ to V_m_^1^ at 650 nm (1.9 eV)) within the van Hove singularities [52]. Here we looked at the high-energy transition at 650 nm to investigate the dispersion of CNT and C-CNT in PI films. The results are shown in Table 4 and Figure 8. The transmittance of CNT/PI and C-CNT/PI films decreased first and then increased with the increase in CNT and C-CNT content. When the filler content was 1%, CNT or C-CNT was not enough to cover the whole film (Figure 5a,b), so the film still maintained a certain transmittance. With the increase in filler content, the transmittance of the films decreased due to the outstanding light absorption of CNT and C-CNT. However, when the content of CNT and C-CNT reached 7 wt.%, the transmittance of the films increased because the high filler content led to a decrease in dispersion. In addition, it is worth noting that the transmittance of C-CNT/PI films was lower than that of CNT/PI films, especially when the filler content was 1 wt.% because the dispersion of C-CNT in the membrane was higher than that of CNT at the same concentration [53]. When CNT or C-CNT agglomerates, the PI film will have areas with few fillers, increasing the light passing through, thus improving the transmittance of the film [54]. The better dispersion of C-CNT than CNT results in lower transmittance of C-CNT/PI films than CNT/PI films and higher transmittance of the film under high filler concentration.

In order to study the dispersion and morphology of CNT and C-CNT in the films, the cross-sections of CNT/PI and C-CNT/PI films with filler contents of 0, 1, 3, 5, 7, and 9 wt.% were observed by SEM (Appendix A). The cross-sections were the fracture surfaces of the broken films after the mechanical properties test. As shown in Figure 9, CNT and C-CNT are marked with red circles. When the film is stretched, the strong adhesive interaction between carbon nanotubes and polymer matrix will produce stress, leading to local plastic deformation and rough fracture surface [47]. The C-CNT was mainly broken near the surface of the PI matrix, and compared with CNT/PI film, C-CNT/PI film had less filler exposed outside the matrix and a shorter filler length. This phenomenon means that the interfacial adhesion between C-CNT and PI matrix was better than CNT, which made C-CNT harder to be pulled out when the film broke [54]. Because of the hydrogen bonds formed by the H atom of the –COOH group on the surface of C-CNT and the C=O bond of the PI molecule, C-CNT was less agglomerated than CNT. As a result, C-CNT can be well wetted by the PI matrix, so the interfacial adhesion between C-CNT and PI matrix was better than CNT. In brief, C-CNT had better dispersion and less aggregation than CNT in the film, so it had better interaction with the PI matrix.

### 3.3. Thermal Properties of CNT/PI and C-CNT/PI Films

The thermal behaviors of PI films with various CNT and C-CNT contents were examined by TGA to evaluate the thermal stability, and the results are shown in Figure 10. The thermal decomposition of the PI film started at about 500 °C [55]. It can be seen from Table 5 that the 5 wt.% loss temperature of the pure PI film was only 551.63 °C. After mixing with CNT or C-CNT, the 5 wt.% loss temperature of the films was significantly increased and reached the highest value when the content was 7 wt.%. With the further increase in the content of CNT or C-CNT, the 5 wt.% loss temperature of the films decreased. This behavior may be due to the fact that excessive fillers tend to agglomerate and hinder heat transfer, generating high-temperature spots, thus affecting the continued improvement of thermal stability. At the same content, C-CNT showed a better effect of improving the thermal stability of the film than CNT, and the 5 wt.% loss temperature could reach up to 595.49 °C (7 wt.% C-CNT/PI film), which may be related to the high thermal stability of C-CNT and good filler-polymer affinity [56]. Moreover, the improved thermal resistance of PI films after mixing with CNT and C-CNT also led to a higher residual rate at 800 °C, from 49.93% (pure PI film) to 59.61% (9 wt.% C-CNT/PI film).

Figure 11 shows the TMA results of PI films with various CNT and C-CNT contents, which reflect the dimensional stability of the films under heating. The temperature displacement curves are nearly linear because the test temperature is well below the glass transition temperature [57]. With the increase in filler content, the CTE of films showed a continuous downward trend and reached the lowest value at 9 wt.%. Because the deformation caused by thermal expansion at −10 °C to 150 °C was very small, the CTE values of CNT/PI film and C-CNT/PI film were very similar at the same content. Therefore, multiple measurements and statistical analyses were performed to obtain accurate results (Table 6). The increase in the CTE value of the film was found when CNT aggregates, so the C-CNT/PI film with the same content had a lower CTE value than the CNT/PI film. This result is consistent with the influence of CNT aggregation in the matrix on the CTE value of the material deduced from theoretical and numerical techniques [58,59]. When the filler content reached 9 wt.%, the CTE of CNT/PI and C-CNT/PI films decreased to 19.74 and 19.04 ppm/K, respectively, and decreased by 25.17% and 27.82% compared with pure PI films. At this time, C-CNT played a better role in reducing CTE because the C-CNT achieved better dispersibility and interfacial properties with the matrix due to the carboxylation modification. The CTE of a composite mainly depends on its structure, and each phase in the composite and the interaction between the phases together determine the CTE. The existence of CNT can be used as a physical barrier to interrupt the crystallization process, thus affecting the size of the lens and the structure of spherulites and ultimately leading to the increase in the amorphous area and the decrease in the crystallinity [22,60]. When the PI film has low crystalline morphology, it will reduce the CTE of PI and cause thermal expansion of the film in the thickness direction [61,62]. Meanwhile, C-CNT had more influence areas on the crystallinity of films (especially at low concentrations) due to its better dispersion in the film than CNT, which led to lower CTE of C-CNT/PI films than CNT/PI films. Moreover, the CTE of CNT and C-CNT is so low that they hardly deform during the experiment, and when the PI matrix is thermally expanded, they will limit the deformation of the matrix as reinforcements. CNT and C-CNT were uniformly dispersed in the PI matrix and had a solid interfacial bonding effect with the matrix, so the CTE of CNT/PI and C-CNT/PI films can be effectively reduced.

### 3.4. Mechanical Properties of CNT/PI and C-CNT/PI Films

In order to investigate the effect of CNT and C-CNT in PI films, the mechanical properties of the films were further tested, and the experimental results are shown in Figure 12. According to the results in Table 7, both the tensile strength and Young’s modulus of films increased up to a maximum value and then decreased with the content of the filler. The tensile strength and Young’s modulus of CNT/PI and C-CNT/PI films reached the highest value at 5 wt.% filler content, and C-CNT had a better mechanical reinforcement effect on PI films. When the content of fillers reached 5 wt.%, the tensile strength of the CNT/PI and C-CNT/PI films increased to 95.61 and 97.52 MPa, respectively, which increased by about 21.50% and 23.93% compared with pure PI films (78.69 MPa). When the content of C-CNT was 5 wt.%, Young’s modulus of C-CNT/PI film reached the highest value of 3.02 GPa, which was 30.82% higher than that of pure PI films (2.31 GPa). However, the turning point of Young’s modulus of CNT/PI film was earlier (3 wt.%), which was only 14.35% higher than that of pure PI films. In particular, compared with CNT/PI films, the tensile strength and Young’s modulus of C-CNT/PI films were higher when the filler content was the same. In Figure 7e,f, the 9 wt.% CNT/PI film surface had small visible particles formed by CNT agglomeration, while the 9 wt.% C-CNT/PI film did not have these particles. Therefore, the dispersion of C-CNT in the films was more uniform than that of CNT, and due to the existence of –COOH groups, both the interaction between C-CNT and PI matrix and the interface performance was improved. The dispersion of fillers in polymers and the interaction between fillers and polymers play a vital role in enhancing the mechanical properties of polymer composites; as a result, the mechanical properties of C-CNT/PI films are better improved. Similarly, when the filler content is too high, the dispersion of the filler in the film will be reduced, and agglomeration will occur. When CNT or C-CNT aggregates, the matrix is difficult to penetrate into it, and the CNT cluster lacks load transfer capability [63]. Meanwhile, there is a poor binding force between CNT clusters [64]; thus, the mechanical properties are mainly determined by the matrix. Therefore, at higher filler content (>5 wt.%), the mechanical properties of CNT/PI and C-CNT/PI films will decrease.

## 4. Conclusions

In conclusion, carboxyl groups were formed on the surface of C-CNT after acid treatment, leading to better dispersion of C-CNT in solution and film than CNT and better interfacial adhesion between C-CNT and PI matrix. Thus C-CNT can better enhance the thermal and mechanical properties of PI films than CNT. C-CNT improved the thermal stability of the films, but high-temperature spots would appear in the films due to high C-CNT content (>7 wt.%), which reduced the thermal stability. The mechanical properties of the films were improved by adding C-CNT. However, it was difficult for C-CNT to be thoroughly wetted by the PI matrix at high content (>5 wt.%), which reduced the load transfer of the matrix to C-CNT, leading to the decline of the mechanical properties of the films. This paper studied the property advantages of C-CNT/PI films and the optimum content required for the highest performance, thus guiding the application of C-CNT/PI films in such fields as aerospace and electronic engineering.

## Figures and Tables

**Figure 1 polymers-14-04565-f001:**
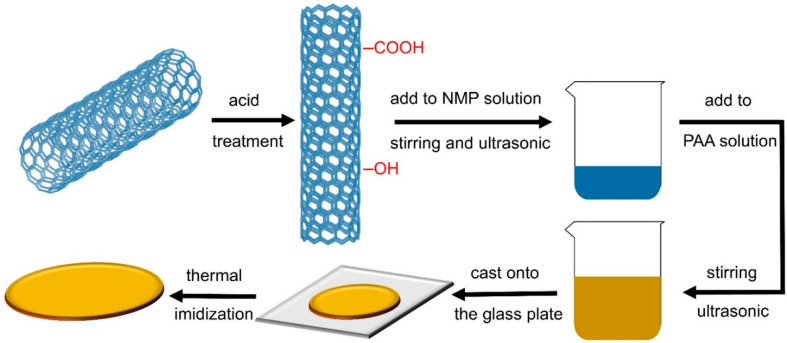
Preparation procedures of films.

**Figure 2 polymers-14-04565-f002:**
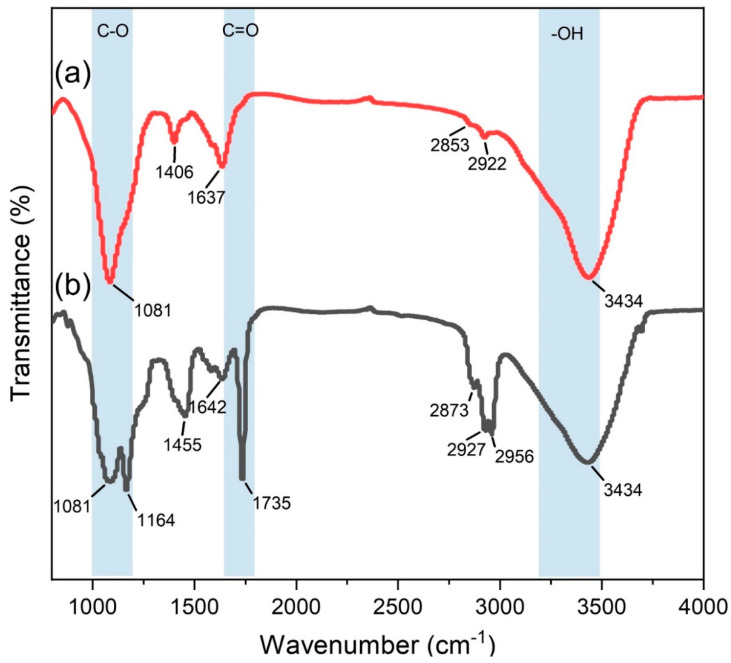
FTIR spectra of (**a**) CNT and (**b**) C-CNT.

**Figure 3 polymers-14-04565-f003:**
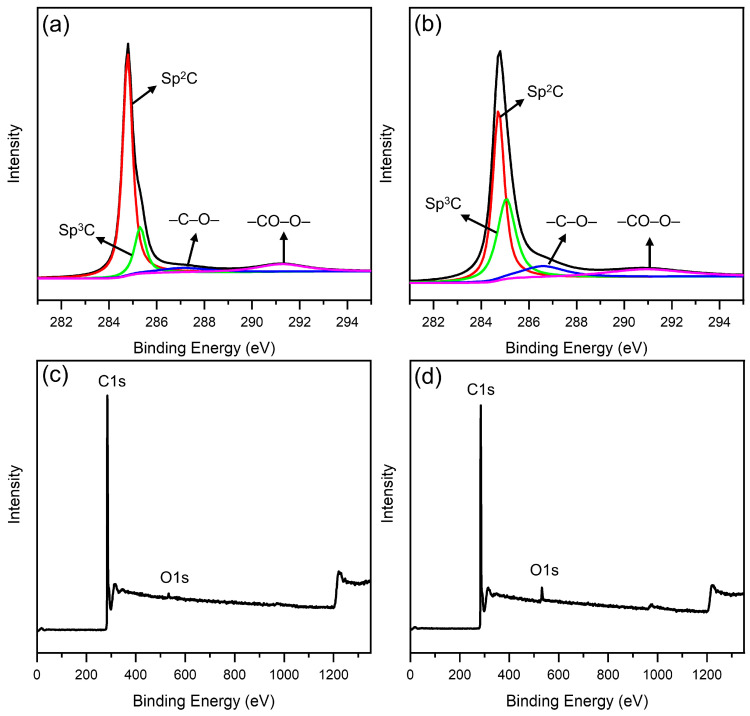
C1s XPS spectra of (**a**) CNT and (**b**) C-CNT, and XPS survey scans of (**c**) CNT and (**d**) C-CNT.

**Figure 4 polymers-14-04565-f004:**
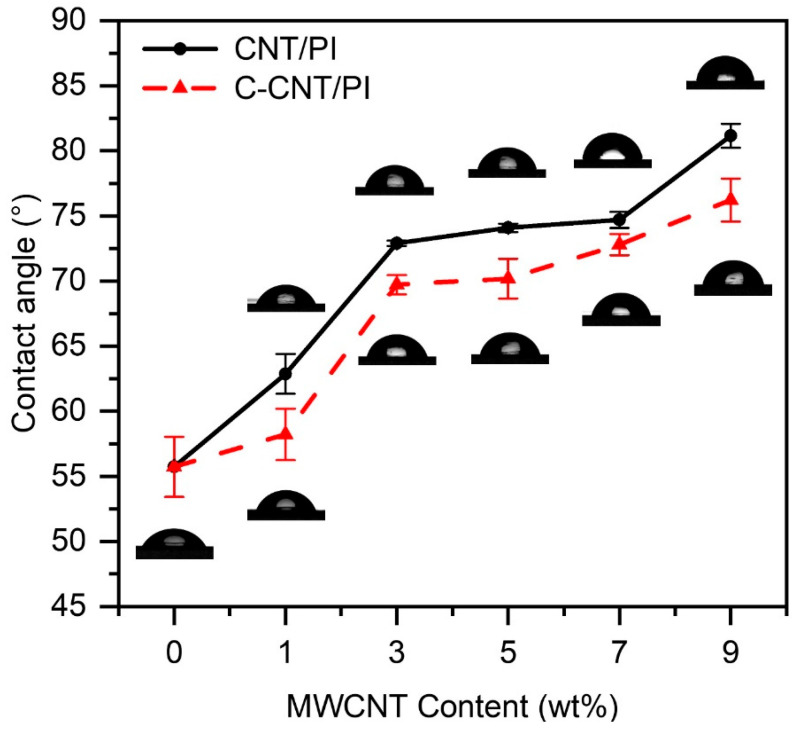
Summary of water contact angle measurements for PI, CNT/PI and C-CNT/PI films.

**Figure 5 polymers-14-04565-f005:**
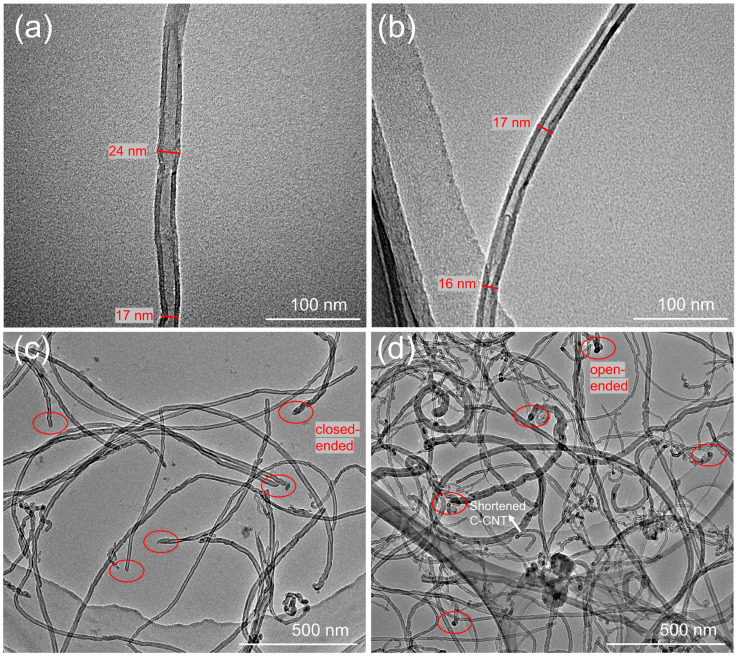
TEM image of (**a**) single CNT, (**b**) single C-CNT, (**c**) CNT with closed ends, (**d**) C-CNT with opened ends.

**Figure 6 polymers-14-04565-f006:**
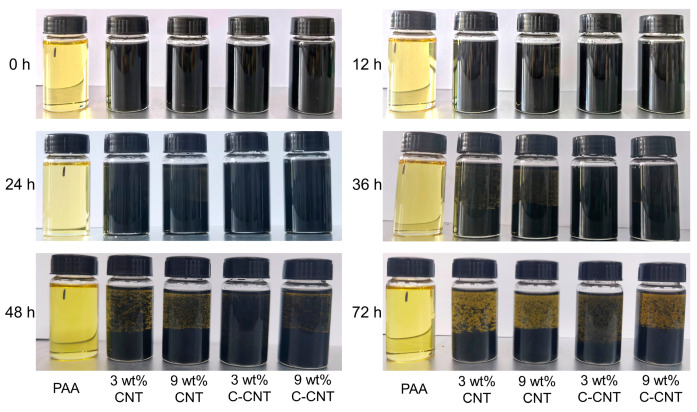
Comparison of photographs of CNT and C-CNT dispersed in solution for different hours.

**Figure 7 polymers-14-04565-f007:**
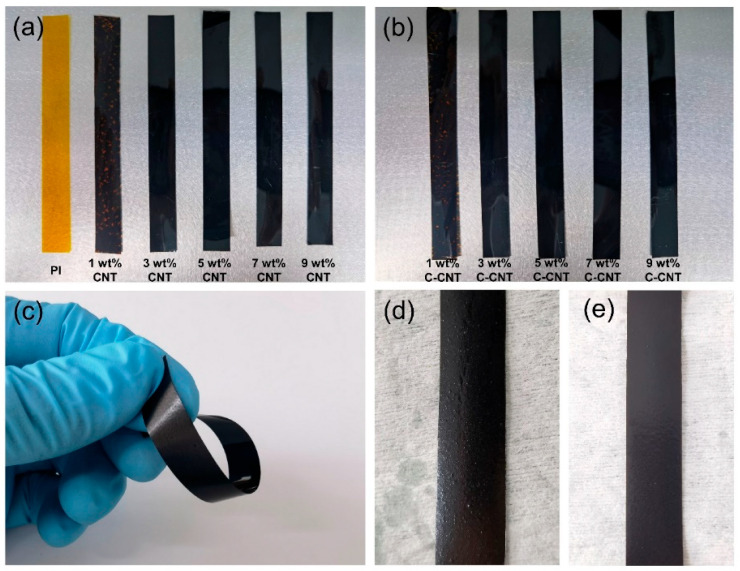
Photographs of PI films with different contents of (**a**) CNT and (**b**) C-CNT, (**c**) the flexible film, (**d**) 9 wt.% CNT/PI film, and (**e**) 9 wt.% C-CNT/PI film.

**Figure 8 polymers-14-04565-f008:**
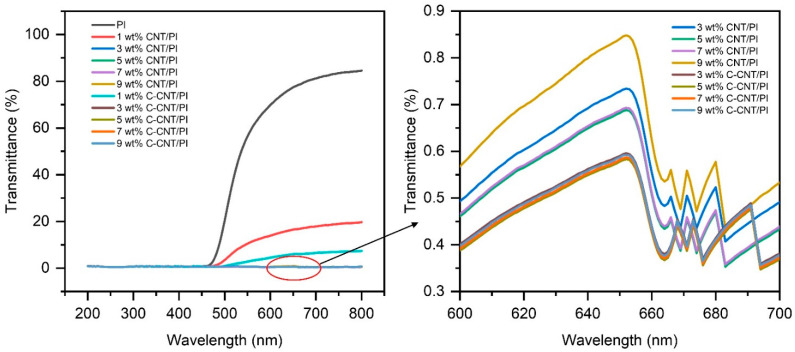
UV–visible spectra of PI, CNT/PI and C-CNT/PI films.

**Figure 9 polymers-14-04565-f009:**
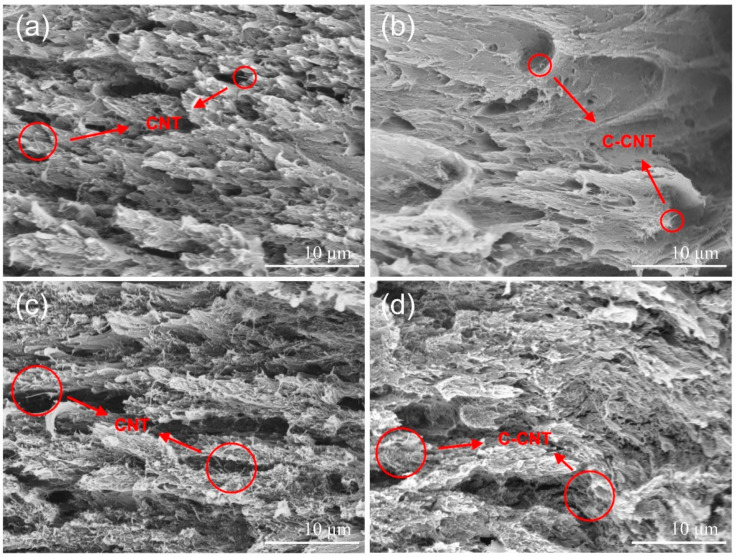
SEM cross-sectional images of (**a**) 1 wt.% CNT/PI film, (**b**) 1 wt.% C-CNT/PI film, (**c**) 9 wt.% CNT/PI film, (**d**) 9 wt.% C-CNT/PI film.

**Figure 10 polymers-14-04565-f010:**
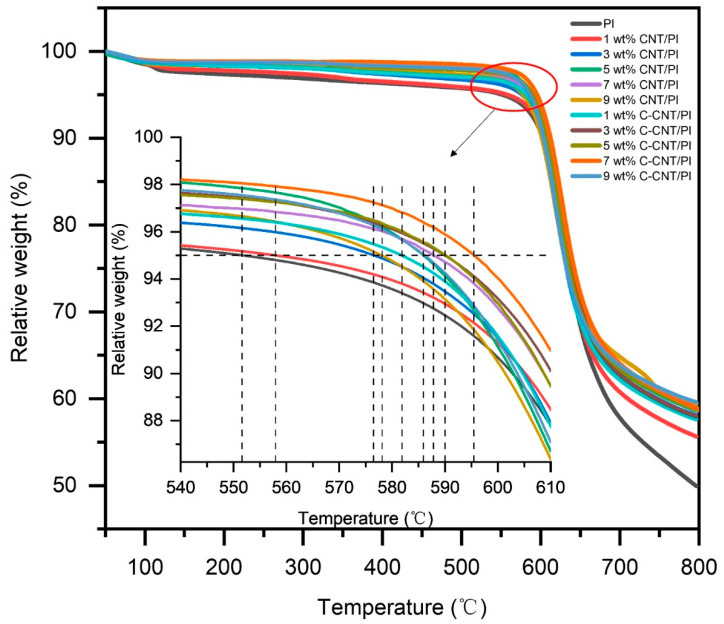
TGA results of PI films with different contents of CNT and C-CNT.

**Figure 11 polymers-14-04565-f011:**
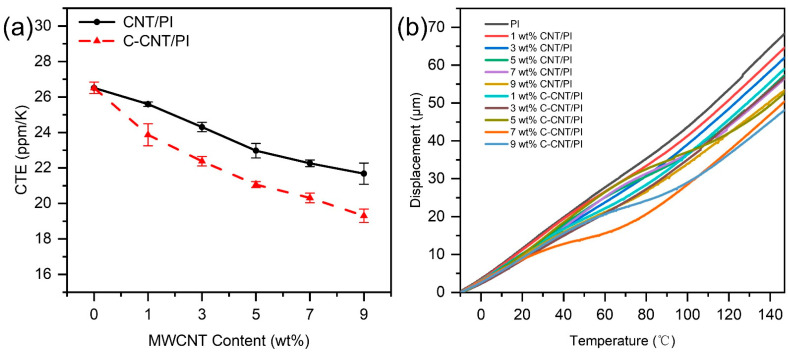
(**a**) Summary of the CTE values of PI films with different contents of CNT and C-CNT and (**b**) TMA curves of PI, CNT/PI and C-CNT/PI films.

**Figure 12 polymers-14-04565-f012:**
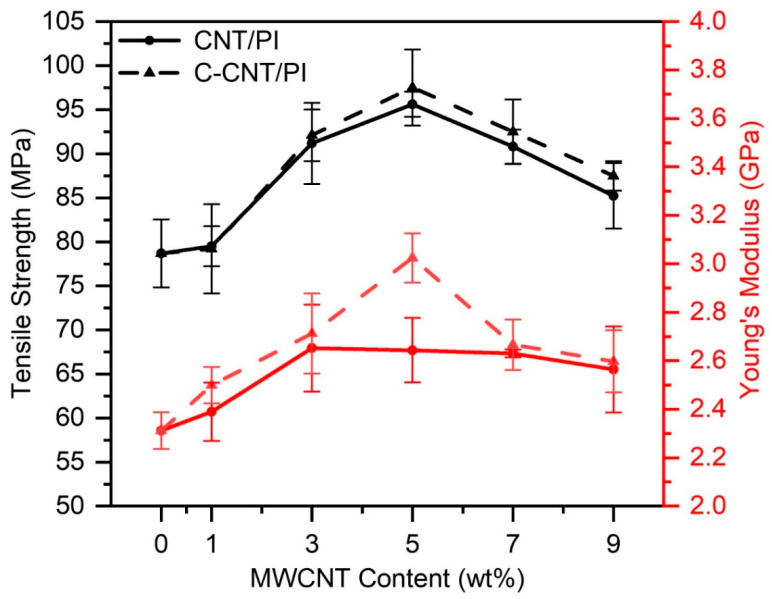
The tensile strength and tensile modulus of PI films with different contents of CNT and C-CNT.

**Table 1 polymers-14-04565-t001:** Content of the functional groups of CNT and C-CNT.

Samples	Contents of the Functional Groups			
	Sp^2^ C (%)	Sp^3^ C (%)	–C–O– (%)	–CO–O– (%)
CNT	65.93	15.70	7.04	11.33
C-CNT	40.46	31.92	11.74	15.88

“CNT” is the abbreviation of carbon nanotube, “C-CNT” is the abbreviation of carboxylated CNT.

**Table 2 polymers-14-04565-t002:** Element content of CNT and C-CNT.

Samples	Element Content (%)				Proportion
	C	O	N	S	O/C
CNT	97.30	1.27	1.28	0.15	0.01
C-CNT	95.20	3.79	0.85	0.15	0.04

“CNT” is the abbreviation of carbon nanotube, “C-CNT” is the abbreviation of carboxylated CNT.

**Table 3 polymers-14-04565-t003:** The water contact angles of PI, CNT/PI and C-CNT/PI films.

Samples	Contact Angle (°)	Samples	Contact Angle (°)
PI	55.74 ± 2.31	-	-
1 wt.% CNT/PI	62.87 ± 1.53	1 wt.% C-CNT/PI	58.23 ± 1.97
3 wt.% CNT/PI	72.90 ± 0.21	3 wt.% C-CNT/PI	69.72 ± 0.74
5 wt.% CNT/PI	74.09 ± 0.32	5 wt.% C-CNT/PI	70.19 ± 1.52
7 wt.% CNT/PI	74.69 ± 0.62	7 wt.% C-CNT/PI	72.79 ± 0.81
9 wt.% CNT/PI	81.17 ± 0.91	9 wt.% C-CNT/PI	76.23 ± 1.65

“PI” is the abbreviation of polyimide, “CNT” is the abbreviation of carbon nanotube, “C-CNT” is the abbreviation of carboxylated CNT; The “±” symbol is used to connect the mean and standard deviation of the measurements.

**Table 4 polymers-14-04565-t004:** Transmittance of PI, CNT/PI and C-CNT/PI films.

Samples	T_650_ (‰)	Samples	T_650_ (‰)
PI	774.18	-	-
1 wt.% CNT/PI	164.98	1 wt.% C-CNT/PI	59.45
3 wt.% CNT/PI	7.28	3 wt.% C-CNT/PI	5.90
5 wt.% CNT/PI	6.82	5 wt.% C-CNT/PI	5.81
7 wt.% CNT/PI	6.83	7 wt.% C-CNT/PI	5.81
9 wt.% CNT/PI	8.41	9 wt.% C-CNT/PI	5.82

“T_650_” is the transmittance at 650 nm, “‰” means one over one thousand.

**Table 5 polymers-14-04565-t005:** The thermal properties of PI, CNT/PI and C-CNT/PI films.

Samples	5 wt.% Loss Temperature (°C)	800 °C Residual Rate (%)
PI	551.63	49.93
1 wt.% CNT/PI	557.97	55.65
3 wt.% CNT/PI	576.47	58.07
5 wt.% CNT/PI	585.93	58.61
7 wt.% CNT/PI	587.85	58.72
9 wt.% CNT/PI	578.11	59.43
1 wt.% C-CNT/PI	581.88	57.64
3 wt.% C-CNT/PI	590.05	57.99
5 wt.% C-CNT/PI	590.09	58.79
7 wt.% C-CNT/PI	595.49	58.93
9 wt.% C-CNT/PI	586.07	59.61

“PI” is the abbreviation of polyimide, “CNT” is the abbreviation of carbon nanotube, “C-CNT” is the abbreviation of carboxylated CNT.

**Table 6 polymers-14-04565-t006:** The CTE values of PI, CNT/PI and C-CNT/PI films.

Samples	CTE (ppm/K)	Samples	CTE (ppm/K)
PI	26.51 ± 0.32	-	-
1 wt.% CNT/PI	25.59 ± 0.12	1 wt.% C-CNT/PI	23.87 ± 0.62
3 wt.% CNT/PI	24.31 ± 0.26	3 wt.% C-CNT/PI	22.38 ± 0.27
5 wt.% CNT/PI	22.97 ± 0.41	5 wt.% C-CNT/PI	21.07 ± 0.17
7 wt.% CNT/PI	22.26 ± 0.18	7 wt.% C-CNT/PI	20.31 ± 0.27
9 wt.% CNT/PI	21.68 ± 0.60	9 wt.% C-CNT/PI	19.30 ± 0.38

“PI” is the abbreviation of polyimide, “CNT” is the abbreviation of carbon nanotube, “C-CNT” is the abbreviation of carboxylated CNT; The “±” symbol is used to connect the mean and standard deviation of the measurements.

**Table 7 polymers-14-04565-t007:** The mechanical properties of PI, CNT/PI and C-CNT/PI films.

Samples	Tensile Strength (MPa)	Tensile Modulus (GPa)
PI	78.69 ± 3.87	2.31 ± 0.08
1 wt.% CNT/PI	79.50 ± 2.27	2.39 ± 0.12
3 wt.% CNT/PI	91.18 ± 4.61	2.65 ± 0.18
5 wt.% CNT/PI	95.61 ± 1.42	2.64 ± 0.13
7 wt.% CNT/PI	90.80 ± 1.92	2.63 ± 0.02
9 wt.% CNT/PI	85.22 ± 3.72	2.56 ± 0.18
1 wt.% C-CNT/PI	79.21 ± 5.06	2.50 ± 0.07
3 wt.% C-CNT/PI	92.10 ± 2.93	2.71 ± 0.17
5 wt.% C-CNT/PI	97.52 ± 4.33	3.02 ± 0.10
7 wt.% C-CNT/PI	92.49 ± 3.68	2.67 ± 0.10
9 wt.% C-CNT/PI	87.50 ± 1.68	2.60 ± 0.13

“PI” is the abbreviation of polyimide, “CNT” is the abbreviation of carbon nanotube, “C-CNT” is the abbreviation of carboxylated CNT; The “±” symbol is used to connect the mean and standard deviation of the measurements.

## Data Availability

The data presented in this study are available on request from the corresponding author.

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
