# Peer review of "Carboxylated Carbon Nanotube/Polyimide Films with Low Thermal Expansion Coefficient and Excellent Mechanical Properties"

_polymers, 2022, doi:10.3390/polym14214565_

Round 1
Reviewer 1 Report
The authors reported on the absorption, thermal and mechanical properties of carboxylated carbon nanotubes in polyimide membranes. The manuscript is well written and organized into logical sections and the results are interesting. There a few minor points in the attached pdf file, requiring the consideration of the authors to improve the manuscript and its presentation.

Author Response
We highly appreciate the comments which help us in improving the quality of this manuscript. According to the comments, we carefully revised the manuscript entitled "Carboxylated carbon nanotubes/polyimide films with low thermal expansion coefficient and excellent mechanical properties" (Polymer-1968067) through the "Track Changes" function. The detailed replies and revisions are listed in the attachment files.

Reviewer 2 Report
Dear authors,
I have reviewed the paper polymers-1968067, entitled "Carboxylated carbon nanotube/polyimide membranes with low thermal expansion coefficient and excellent mechanical properties". In this study, the authors produced PI membranes with the addition of 1, 3, 5, 7, and 9 wt.% CNT and C-CNT. Mechanical and physical-chemical properties and thermal stability of the membranes were assessed. Overall, he addition of C-CNT at moderate amounts (3 wt%) had the best effect in improving membrane properties. The study is interesting and has merit, but some issues must be addressed before reconsideration for publication.
- This study apparently developed films, and not membranes since no possible use as a membrane was presented and neither discussed in this paper. I suggest replacing the term 'membranes' with 'films' throughout the paper.
- The authors should carry out some statistical analyses to formally determine which filler (CNT/C-CNT) and which concentration (zero, 1, 3, 5, 7, and 9 wt.%) had the best performance regarding the evaluated parameters (physical-chemical, mechanical, thermal).
- If possible, the authors could determine the contact angles to verify the effect of the filler and the concentration on the surface hydrophobicity/hydrophilicity of the films. This is very important depending on the application desired.
- A table covering all results with standard deviations of the mechanical analysis could be provided. This will help the reader to visualize the results and help the authors to discuss the paper.
- In the presentation of several results only the filler concentrations of zero, 1, 3, and 9 wt.% were shown. I recommend presenting all concentrations studied (1, 3, 5, 7, and 9 wt%).
Please refer to the attached pdf file for specific recommendations.

Author Response

(The authors gave the same response as above.)

Round 2
Reviewer 2 Report
Dear authors,
I reviewed the new version of the article and see that the recommendations and suggestions have been properly addressed. In my opinion, the paper was much improved, and I believe it can be published in the present form.